# Identification of Chromosomal Regions and Candidate Genes for *Round leaf* Locus in *Cucumis melo* L.

**DOI:** 10.3390/plants13081134

**Published:** 2024-04-18

**Authors:** Xufeng Fang, Zicheng Zhu, Junyan Li, Xuezheng Wang, Chunhua Wei, Xian Zhang, Zuyun Dai, Shi Liu, Feishi Luan

**Affiliations:** 1Key Laboratory of Biology and Genetic Improvement of Horticulture Crops (Northeast Region), Ministry of Agriculture and Rural Affairs, College of Horticulture and Landscape Architecture, Northeast Agricultural University, Harbin 150030, China; 18800469189@163.com (X.F.); zzc1983sc@163.com (Z.Z.); l2580868751@163.com (J.L.); xz6206815@163.com (X.W.); 2College of Horticulture and Landscape Architecture, Northeast Agricultural University, Harbin 150030, China; 3College of Horticulture, Northwest A&F University, Xianyang 712100, China; xjwend020405@nwafu.edu.cn (C.W.); zhangxian@nwafu.edu.cn (X.Z.); 4Anhui Jianghuai Horticulture Technology Co., Ltd., Hefei 230031, China; daizuyun@jhseed.com

**Keywords:** melon, leaf morphology, genetic mapping, molecular markers

## Abstract

Leaf morphology plays a crucial role in plant classification and provides a significant model for studying plant diversity while directly impacting photosynthetic efficiency. In the case of melons, leaf shape not only influences production and classification but also represents a key genetic trait that requires further exploration. In this study, we utilized forward genetics to pinpoint a recessive locus, dubbed *Cmrl* (*Round leaf*), which is responsible for regulating melon leaf shape. Through bulked segregant analysis sequencing and extensive evaluation of a two-year F_2_ population, we successfully mapped the *Cmrl* locus to a 537.07 kb region on chromosome 8 of the melon genome. Subsequent genetic fine-mapping efforts, leveraging a larger F_2_ population encompassing 1322 plants and incorporating F_2:3_ phenotypic data, further refined the locus to an 80.27 kb interval housing five candidate genes. Promoter analysis and coding sequence cloning confirmed that one of these candidates, *MELO3C019152.2* (*Cmppr* encoding a pentatricopeptide repeat-containing family protein, Cmppr), stands out as a strong candidate gene for the *Cmrl* locus. Notably, comparisons of *Cmrl* expressions across various stages of leaf development and different leaf regions suggest a pivotal role of *Cmrl* in the morphogenesis of melon leaves.

## 1. Introduction

Leaves are important source organs of plants. Their shapes and sizes directly affect photosynthetic and transpiration efficiency, and they are also important factors in the establishment of plant morphology [1]. The study of the morphological variation of leaves is of great significance for the understanding plant phylogeny, vegetation resources and distribution, and for promoting biodiversity conservation [2]. The above-ground structures of higher plants are differentiated from the shoot apical meristem (SAM), which can be divided into central and peripheral regions according to their potentials for cellular development [3,4,5]. Cells of the central region are undifferentiated and maintain totipotency, while peripheral cells surround the central region and differentiate into tissues such as leaves, branches and floral organs [6,7,8]. Moreover, after the establishment of polarity, leaves gradually develop highly diverse geometric structures of marginal morphology that include full margin, serrated, lobed, and deeply lobed variants [9]. Plants exhibit highly adapted leaf shapes that ensure survival in particular environments. For example, deep lobes facilitate heat diffusion and reduce leaf temperature, thereby reducing light burn [10]. Compared with deep-lobed leaves, round leaves are capable of higher light interception. The lower thermal conductivity of leaf edges and higher leaf surface temperatures improve photosynthesis and dark respiration efficiency [11,12]. Deeply lobed leaves may adapt to cold temperatures by increasing the internal flow of liquid and promoting gas exchange and carbon fixation between the leaves and the external environment, thus compensating for the inhibition of photosynthesis by low temperatures [2,13]. In addition, water spitting can release excessive water flow between mesophyll cells, thereby alleviating excessive root pressure and promoting resistance to drought [14,15]. The origins of leaf morphological diversity are complex, and excluding environmental factors, they are primarily influenced by genetic regulatory networks that differentiate the stem apical meristem into varied leaf morphologies [16].

Pentatricopeptide repeat (PPR) proteins constitute a family of nucleic acid binding proteins consisting of poorly conserved tandems comprised of 2 to 26 copies of 31–36 amino acid motif repeats [17]. PPR proteins can be grouped according to the characteristics of their structural domains with the typical P subfamily containing only P motifs and the PLS subfamily containing varying lengths [18]. The PLS subfamily features an aspartate–tyrosine–tryptophan structural domain attached at the N-terminus, which is divided into four subclasses (PLS, E, E+, and DYW) based on the C-terminal structural domain [19]. The PPR proteins represent one of the largest protein families of plants, and they play a broad and crucial role in growth and development [20]. PPR proteins were first discovered during a systematic screening of mitochondrial and chloroplast proteins in the model plant *Arabidopsis* [17]. They are encoded by nuclear genes but are translated in plasmids and mitochondria. While extensive research has been conducted on the PPR gene family in model plants, non-model plant species have garnered significantly less attention [21]. In *Arabidopsis*, the DYW-type PPR proteins were associated with leaf development. YS1, which has a DYW motif at its C-terminus, is a PPR protein localized in chloroplasts. The activity of the plastid-encoded polymerase encoded by the *ys1* mutant decreases as light and capacity decline; consequently, the leaves appear yellow and the leaf area is smaller than that of the wild type [22,23]. AtECB2 also has a DYW domain at its C-terminus and is localized to chloroplasts; mutation causes chloroplast abnormalities and leaf albinism [24]. In addition to the DYW-type PPR proteins belonging to the PLS subfamily, PPR proteins of the P subfamily have also been associated with leaf morphogenesis. *AtSLO3* encodes a PPR protein of the P subfamily; a mutant resulted in curling and ruffling rosette leaves. *AtSLO3* may participate in the auxin metabolic pathway, thus affecting leaf shape [25]. *Atppr596* can edit mitochondrial transcripts and is another P subfamily PPR protein in *Arabidopsis*. The leaves of an *Atppr596* mutant were smaller in the early growth stage. Mutant plants were also much smaller than wild type; by the late growth stage, the size of *Atppr596* mutant and wild-type plants were similar, but the leaves of the *Atppr596* mutant were more curled and irregularly shaped [26].

The genetic regulation of leaf shape in cucurbit crops has been unraveled. Wei et al. localized the monodominant *LOBED LEAF 1* locus within the 127.6 kb interval on watermelon chromosome 4 and hypothesized that *ORF18* and *ORF22* were the most likely candidate genes for the *ClLL1* locus [27]. Subsequently, Xu et al. shortened the locus to within the 98.23 kb interval by constructing different populations. In addition, the homolog of *LATE MERISTEM IDENTITY1* (*LMI1*), *Cla97C04G076510*, was identified as a candidate gene for the *non-lobed leaf* (*Clnll*) locus [28]. In zucchini, *CpDll* encoded a homolog of HD-Zip I transcription factor and was localized in a 21 kb interval on chromosome 10, regulating the formation of deeply lobed leaves [29]. However, few studies have addressed melon leaf shapes. The only study reported to date examined the *palmate lobe locus* (*pll*) on melon linkage group III [30]. However, no further investigations have been conducted to confirm the findings of that study. The exploration of leaf morphogenesis at a genetic level not only facilitates the identification of the genes that control leaf shape but also offers insights into the intricate regulatory mechanisms involved. In this study, we aimed to determine the genes that may potentially regulate melon leaf morphology and attempted to provide genetic resources to elucidate their transcriptional regulation. We discovered the genetic inheritance of the *Cmrl* (*Round leaf*) locus by crossing the round-leaved MR-1 with the deeply lobed-leaved PI 614174, and we finely mapped and cloned a target gene by bulk segregant analysis sequencing (BSA-seq) and F_2_ population genotypes in combination with the F_2:3_ family phenotypes. The cloning of coding regions and promoters combined with studies of gene expressions in different leaf sites and at different stages of development indicate that *Cmrl* plays a key role in melon leaf morphogenesis.

## 2. Results

### 2.1. The Round leaf in MR-1 Is Controlled by a Single Recessive Locus

All F1 progeny of round-leaved MR-1 crossed with deeply lobed-leaved PI 614174 displayed deeply lobed leaves (Figure 1 and Appendix A). Of the 220 F_2_ individuals planted in the spring of 2021, 169 plants showed deeply lobed leaves and 51 exhibited round leaves, which conformed to a 3:1 genetic ratio (χ^2^ = 0.209, *p* = 0.648, Figure 1, Table 1). In another F_2_ population derived from the same parental lines and comprising 1155 plants planted in the fall of the same year, 853 plants showed deeply lobed leaves, and the remaining 302 plants displayed round leaves, which also conformed to the 3:1 ratio (χ^2^ = 0.43, *p* = 0.512, Figure 1, Table 1). All the above results indicated that the Cmrl locus in melon is regulated by a single recessive gene.

### 2.2. Mapping of the Cmrl Locus into an 80.27-kb Region

BSA-seq included 20 individuals from the F_2_ population (1155 individuals) with extreme traits of round or deeply lobed leaves, forming round leaf and deeply lobed leaf pools, respectively, and they also included two parental lines with resequencing data (MR-1 and PI 614174). For MR-1 and PI 614174, 10.79 Gb (77,308,457, 95.71% mapped ratio) and 9.88 Gb (70,750,956, 97.94% mapped ratio) clean reads were obtained, respectively. A total of 70,864,706 and 70,700,790 clean reads were acquired from the deeply lobed leaf pool (96.83% mapped ratio) and round leaf pool (96.98% mapped ratio) with Q30 quality scores of 92.77% and 92.41%, respectively. To visualize the *Cmrl* locus region, the SNP-index values of the two extreme pools were subtracted based on 95% confidence intervals and correlated to the melon reference genome (DHL92 v3.6.1). We subsequently mapped the *Cmrl* locus to chromosome 8 within the interval of 11.88 to 19.57 Mb according to the DHL92 v3.6.1 genome (~7.69 Mb, Figure 2a).

To validate the BSA-seq interval, we developed 15 markers with polymorphisms on chromosome 8, including 13 CAPS markers and 2 InDel markers. These markers were genotyped in two F_2_ populations (220 plants and 1155 plants, respectively) planted in 2021, and they facilitated the preliminary location of the *Cmrl* locus into an approximately 537.07 kb interval (Chr08: 14,137,876 to 14,674,941) between markers *Chr8_14137876* and *Chr8_14674941* with one and two recombinational events, respectively (Figure 2b). 

To further shorten the candidate region of the *Cmrl* locus, we genotyped another F_2_ population (1322 plants, derived from the same parental lines) with markers *Chr8_14137876* and *Chr8_14674941*. Between the two flanking markers, we designed 10 new markers with polymorphisms, including six CAPS markers, one InDel marker, and three KASP markers, in combination with F_2:3_ family line phenotypes. Finally, we fine-mapped the *Cmrl* locus to the 80.27 kb interval between markers *Chr8_14574340* and *Chr8_14654609* with two and seven recombinational events, respectively (Figure 2c).

### 2.3. Pentatricopeptide Repeat-Containing Family Protein Is the Candidate Gene for Cmrl Locus

Based on the DHL92 v3.6.1 version of the melon reference genome, a total of five candidate genes were annotated in the fine localization interval (Appendix A). We first performed a preliminary analysis of variability in the sequences of the coding regions of the five candidate genes based on the resequencing data of the two parental lines and verified by gene cloning, and we found that none of the coding regions of *MELO3C019151.2*, *MELO3C033243.2*, *MELO3C019153.2*, and *MELO3C019154.2* contained non-synonymous mutations or structure variations. *MELO3C019152.2* had a non-synonymous SNP ^14,624,133^ mutation (G → A, at exon 14,624,133 bp) between the two parental lines in the coding region. This mutation changed the amino acid sequence from glycine (Gly) to arginine (Arg, Figure 3). To further verify the stability of SNP ^14,624,133^, the coding regions of 10 additional melon materials with different leaf morphologies were cloned. SNP ^14,624,133^ was presented consistently as G (coded glycine) with MR-1 in all four melons with round leaves, while the remaining six deeply lobed leaf materials together with PI 614174 were presented as A (coded arginine), suggesting that SNP ^14,624,133^ is a stable specific mutation (Figure 3). Subsequently, the promoter sequences (start codon ATG upstream 1919 bp) of *MELO3C019152.2* in both parental lines were cloned. Sequence comparison detected no mutations in *cis*-acting elements or structural variations in the promoter region. Interestingly, we found four potential TGAC core *cis*-acting elements at the promoter of *MELO3C019152.2* (Appendix A).

We then compared the relative expressions of the five candidate genes during leaf development between the two parental lines. Except for *MELO3C019151.2* and *MELO3C019152.2*, the relative expressions of candidate genes in MR-1 and PI 614174 were similar at all stages of leaf development. The expression levels of *MELO3C01951.2* and *MELO3C019152.2* in deeply lobed leaved PI 614174 were significantly higher than in round leaf materials (Figure 4). Our analysis of the spatiotemporal expression of *MELO3C019152.2* in specimens from the nine corresponding leaf sites of both parental lines at 50 DAS (days after sowing) described above in Section 4.5 (Figure 5a) disclosed that in the round-leaved MR-1, *MELO3C019152.2* expression was similar throughout the leaf; however, in the deeply lobed-leaved PI 614174, the expression level of *MELO3C019152.2* was significantly higher on the concave surface than at the apical leaf margin, suggesting that the abundance of *MELO3C019152.2* plays an important role in lobed leaf formation (Figure 5b).

### 2.4. Cmrl Encodes a DYW-PPR Protein

We found that *Cmrl* has no introns and that the exon (full-length 2127 bp, Figure 3) encodes a sequence of 709 amino acids that constitute 15 PPR-repeat motifs. The transcript contained E1, E2, and DYW structural domains at its N-terminus, and it belongs to the typical DYW type of the PLS subfamily (Figure 6a). A comparison of *Cmrl* homologs in various species such as watermelon, pumpkin, rice, tomato, grape, and *Arabidopsis thaliana* revealed that all contain conserved PPR-motifs and DYW domains (Figure 6b). Laser confocal microscopy showed that *Cmrl*-GFP co-localized with chloroplast fluorescence in *N. benthamiana* leaf cells, suggesting that *Cmrl* is localized in chloroplasts (Figure 6c).

## 3. Discussion

The present study revealed that the round leaf trait is controlled by a single recessive gene. This finding is similar to those of studies of leaf phenotypes of other cucurbit crops that demonstrated that *lo-1* (single recessive gene) and domain allele *Lo-2* in *Cucurbita maxima* [31,32], *CpDll* in *Cucurbita pepo* L. [29], *ClLL1*, and *ClNll* regulate the lobed leaves of watermelon [27,28]. However, in contrast to the findings of the present study, Gao et al. located the gene (*pll*) regulating the *palmately lobed leaf* in BM7 on melon linkage group III, which is reasonable because the palmate lobed leaf is regulated by a recessive gene [30]. Notably, Wang et al. located the melon leaf shape locus in the interval 14,602,613–14,661,811 bp (DHL92 v3.5.1) on chromosome 8 by a GWAS (genome-wide association study) analysis of 2083 melon materials [33]. Similarly to the findings of the present study, *Cmrl* was located in the physical interval 14,574,340–14,654,609 on chromosome 8 (Figure 2).

Primary and fine mapping suggested that *MELO3C019152.2* is a candidate gene for the *Cmrl* locus, which we validated by the following three lines of evidence. First, gene cloning revealed that only the coding region of *MELO3C019152.2* had the nonsynonymous mutation SNP ^14,624,133^ in both parental lines (Figure 3). Second, 10 natural materials verified that SNP ^14,624,133^ co-segregated with melon leaf morphology and was a stable mutation (Figure 3). Third, qRT-PCR at different stages of leaf development showed that *MELO3C019151.2* and *MELO3C019152.2* were highly expressed in deeply lobed-leaved PI 614174, and that its expression gradually decreased during leaf development (Figure 4). Discrimination of these two genes based solely on their expressions during different periods of leaf development is difficult. However, spatiotemporal expressions in different leaf blade sites suggest that *MELO3C019152.2* is essential in the formation of deeply lobed leaves (Figure 5). All three of these lines of evidence confirmed *MELO3C019152.2* as the most likely candidate gene for the *Cmrl* locus. Notably, *MELO3C019151.2* showed a similar expression trend at different periods of leaf development; consequently, verification by ontogenetic transformation will be necessary for full elucidation. 

The leaf lobe is synergistically regulated by genes other than those that encode PPR proteins, such as those of the knotted 1-like homeobox (KNOX) transcription factor family [34]. *KNOX* possesses homologous heterotypic structural domains, and it belongs to the three-amino-acid-loop-extension (*TALE*) family, which plays important roles in the maintenance of cellular totipotency and the regulation of leaf development [35]. KNOX family transcription factors underlie lobed leaf morphogenesis in multiple plants such as *Lilium tsingtauense* [36], citrus [37], barley [38], and *Arabidopsis* [35,39,40]. In lettuce, LsKN1 not only binds the promoter of *LsPID* to up-regulate the synthesis of growth hormone but also binds specifically to the *LsAS1* promoter to down-regulate *LsAS1* expression and synergistically regulate the formation of palmately lobed leaves [41,42]. The heterologous overexpression of a DYW-type PPR protein VvPPR1 in *Arabidopsis* caused leaf curling, while expressions of both *AtKAN2* and *AtKNAT6* were up-regulated, suggesting that a reciprocal relationship between *VvPPR1* and the KNOX transcription factor plays an important role in grape leaf morphogenesis [43]. In addition, the *KNOX* encoded transcription factor exerts its function by binding a core *cis*-element containing TGAC [44,45,46]. Coincidently, this study disclosed four TGAC-containing *cis*-acting elements in the *Cmrl* promoter (Appendix A). We predicted the upstream transcription factors and binding sites of *Cmrl* using PlantTFDB (http://plantregmap.gao-lab.org/binding_site_prediction.php, accessed on 13 December 2023). We found that KNOX transcription factors of the *TALE* family were present upstream of *Cmrl*. Hence, we hypothesized that KNOX regulates *Cmrl* expression by binding to the TGAC core element. Subsequent transcriptional regulation studies will investigate whether there is an interplay between CmKNOX and *Cmrl*. 

## 4. Materials and Methods

### 4.1. Plant Materials

Round-leaved melon MR-1 was used as the female, while the deeply lobed-leaved melon PI 614174 was used as the male parent. PI 614174 is a wild melon introduced from the U.S. National Plant Germplasm System, and MR-1 is preserved by our laboratory after generations of self-crossing. All parental lines and F1 were planted in the Xiangyang Experimental Agricultural Farm of Northeast Agricultural University and were strictly hand-pollinated during the flowering period. For the F_2_ population, 220 plants were first planted in the spring of 2021 for investigating the inheritance pattern of the round leaf trait, which was followed by 1155 F_2_ plants planted in the fall, combining the spring and fall phenotypes and genotypes to initially excavate the *Cmrl* locus. In spring 2022, an expanded planting of the F_2_ population (1322 plants) was used to screen for recombinant plant selection and subsequent fine mapping.

### 4.2. BSA-Seq and Initial Mapping

Twenty individuals exhibiting the extreme traits of either round or deeply lobed leaves from the F_2_ population were selected. Equal amounts of genomic DNA were extracted and mixed to construct two gene pools encoding either round or deeply lobed leaves. Together with the genomic DNA of the two parental lines, the four samples were re-sequenced using the Illumina HiSeq Xten platform (20× coverage) at the BGI Research Institute (Shenzhen, China). The resequencing raw data were filtered to obtain clean data, after which the two gene pools were compared with the melon reference genome (http://cucurbitgenomics.org/ftp/genome/melon/v3.6.1/, accessed on 5 January 2022) by using Burrows-Wheeler Aligner software (https://bio-bwa.sourceforge.net/, accessed on 7 January 2022) [47]. Chromosomal regions exceeding the thresholds (at 95% confidence levels) were considered to be associated with the target trait.

To confirm candidate regions based on BSA-seq, resequencing-based polymorphic InDel (insertion–deletion), CAPS (cleaved amplified polymorphic sequences), and KASP (Kompetitive allele-specific PCR) markers were developed to genotype the F_2_ generation (Appendix A). Sequences of ~500 bp upstream and downstream of the SNP site were extracted, and the candidate SNP sequences were screened for CAPS sites using SNP2CAPS software (https://pgrc.ipk-gatersleben.de/snp2caps/, accessed on 12 January 2022 ) [48] in combination with restriction endonuclease cleavage site information and converted into CAPS-tagged SNP sites for primer design. Based on the parental InDel information, sequences containing 100 bp upstream and downstream of the InDel site were intercepted, and deletion/insertion sites with the number of differential bases between 3 and 10 bp in the resequencing data were selected. InDel labeling was developed using Primer Premier 5.0 software [48]. For SNP loci that could not be converted into CAPS/InDel markers, LGC-KASP primers were designed and sent to the Vegetable Research Center of Beijing Academy of Agriculture and Forestry for genotyping. Round-leaved plants from two F_2_ populations planted in 2021 were used for primary mapping of the *Cmrl* locus.

### 4.3. Fine Mapping

A total of 1322 F_2_ plants were genotyped for two flanking markers in the initial region and screened for recombinant individuals. Ten new polymorphic markers were developed based on the initial mapping, including 6 CAPS markers, 1 InDel marker, and 3 KASP markers (Appendix A) for recombinant genotyping to detect more recombination events. The genotypes of recombinants with dominant traits (deeply lobed leaves) were identified by the leaf shape segregation of their F_2:3_ family lines in the fall of 2022 with a minimum of 25 plants per each family.

### 4.4. Gene Annotation and Cloning

Candidate genes in the fine mapping regions were annotated with the melon reference genome DHL92 v3.6.1 (http://cucurbitgenomics.org/organism/18, accessed on 19 February 2023). Coding regions and promoter sequences of candidate genes were first compared with the resequencing data of the two parental lines and verified by sequence cloning. Subsequently, cloning of the coding regions of another 10 melon materials with different leaf morphologies was conducted to detect whether the mutation sites were common and related to leaf shape among the melon nature panel (Appendix A). For the promoter sequences, *cis*-acting elements were first analyzed by Plant CARE (http://bioinformatics.psb.ugent.be/webtools/plantcare/html/, accessed on 13 December 2023) after cloning, and then potential transcription factors and their binding sites were predicted using online Plant TFDB tools (http://plantregmap.gao-lab.org/binding_site_prediction.php, accessed on 13 December 2023).

### 4.5. RNA Extraction and Gene Expression Analysis

RNA of the two parental lines was extracted from leaves at 25, 35, and 50 DAS using TransZol UP (TransGen Biotech, Beijing, China). In addition, total RNA was extracted from 9 corresponding sites of PI 614174 and MR-1 leaf tissues at 50 DAS, of which 2, 4, 6, and 8 were on the concave surfaces of the leaf blades, and 1, 3, 5, 7 and 9 were on the apical margins (Figure 5a). Next, cDNA was synthesized using cDNA Synthesis SuperMix (TransGen Biotech). The nucleotide sequences of the qRT-PCR primers are given in Appendix A. *Actin* was referred to *MELO3C008032.2* from the cucurbit database melon genome (DHL92 v3.6.1). qRT-PCR assays were performed using a Bio-Rad Thermal Cycler (Hercules, CA, USA), and relative gene expressions were calculated using the 2^−ΔΔCT^ method [49]. Samples of MR-1 at 25 DAS were used for calibration.

### 4.6. Subcellular Localization

To analyze subcellular localization, we cloned the sequence of the *Cmrl* coding region without the stop codon into the pAN580-GFP fusion expression vector (Appendix A). The constructs were introduced into GV3101 with the helper plasmid pSoup19 and transiently transformed into 5-week-old leaves of *N. benthamiana*. The leaves were incubated under low light conditions in a light incubator with a photoperiod of 16 h light/8 h dark for 48–72 h and then observed using a laser confocal microscope (TCS SP8, Leica, Berlin, Germany).

### 4.7. Statistical Analysis

The data statistics of the charts were obtained using GraphPad Prism 8.0 software (La Jolla, CA, USA), and all figures were drawn using AI (Adobe Illustrator, San Jose, CA, USA). Statistically significant differences were determined by using one-way ANOVA with Tukey’s honestly significant difference. Different letters on the figures indicate statistically significant differences (*p* < 0.05).

## 5. Conclusions

To determine the latent genes regulating melon leaf morphology, we crossed the round-leaved MR-1 with the deeply lobed-leaved PI 614174, in which the inheritance of the *Cmrl* locus is regulated by a single recessive gene. We subsequently finely mapped a candidate gene by BSA-seq and F_2_ population (1375 and 1322 plants) genotypes in combination with the F_2:3_ family phenotypes. *Cmrl* encoded a DYW-type PPR protein that is highly expressed in the recesses of deeply lobed leaves. These results provide new genetic resources for understanding the molecular mechanisms by which the *Cmrl* locus regulates melon leaf morphology.

## Figures and Tables

**Figure 1 plants-13-01134-f001:**
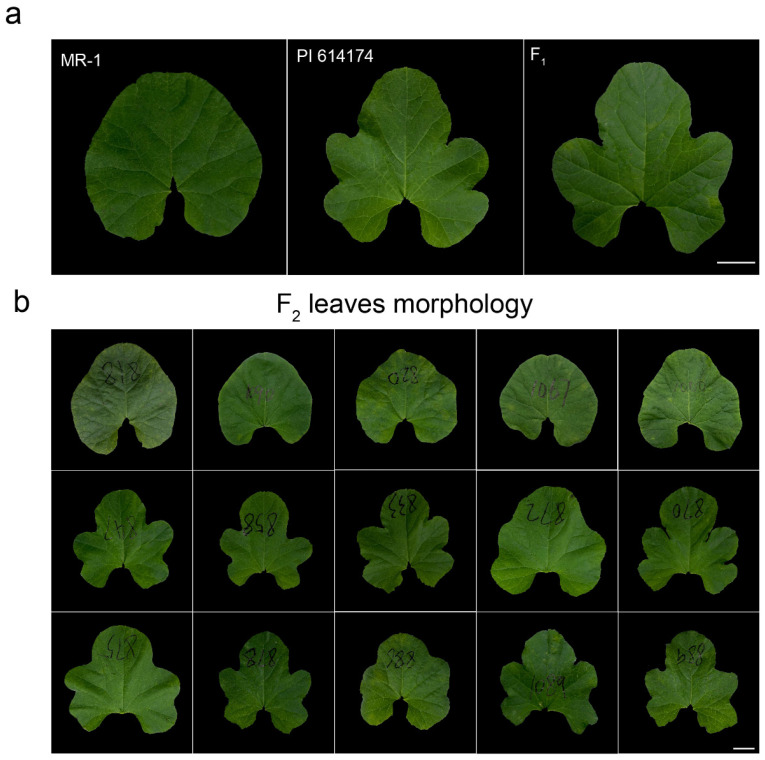
Separation of leaf morphology in the two parental lines and different generations. (**a**) Left to right, leaf morphology of MR-1, PI 614174 and F_1_. Bar = 2.5 cm. (**b**) Leaf morphology of F_2_ generations. Bar = 2.5 cm.

**Figure 2 plants-13-01134-f002:**
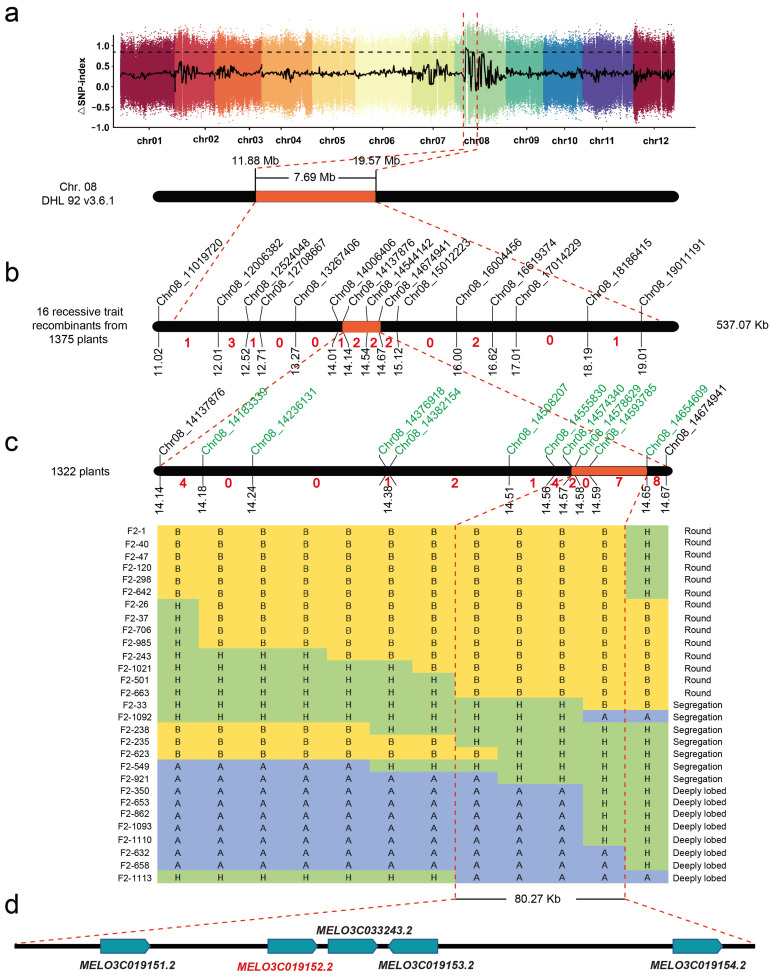
Genetic mapping of the *Cmrl* locus. (**a**) BSA-seq analysis of leaf shape. The *x*-axis denotes the uniformly distributed 12 chromosomes of melon, the *y*-axis represents the mean value of delta SNP between the round and deeply lobed leaf extreme pools. The black dashed line indicates the fitted delta SNP index with a threshold of 95% under the permutation test with B = 1000 (number of random permutation sampling). BSA-seq initially located the *Cmrl* locus within the interval of ~11.88–19.57 Mb on chromosome 8 (DHL92 v3.6.1). (**b**) Initial mapping of the *Cmrl* locus. Sixteen recessive recombinant plants were selected from 1375 F_2_ individuals with final anchoring intervals (~537.07 kb) between *Chr8_14137876* and *Chr8_14674941*. The red numbers between two markers indicate the number of individual recombinant plants. (**c**) Fine targeting of the *Cmrl* locus. The *Cmrl* locus was fine-mapped into the 80.27 kb interval between markers *Chr8_14574340* and *Chr8_14654609* by using a large population (1322 plants) combined with recombinant plant screening. (**d**) Five candidate genes are hidden within the fine mapping interval, which are respectively annotated as *MELO3C019151.2*, *MELO3C019152.2* (*Cmppr*), *MELO3C033243.2*, *MELO3C019153.2* and *MELO3C019154.2* according to the melon reference genome (DHL92 v3.6.1).

**Figure 3 plants-13-01134-f003:**
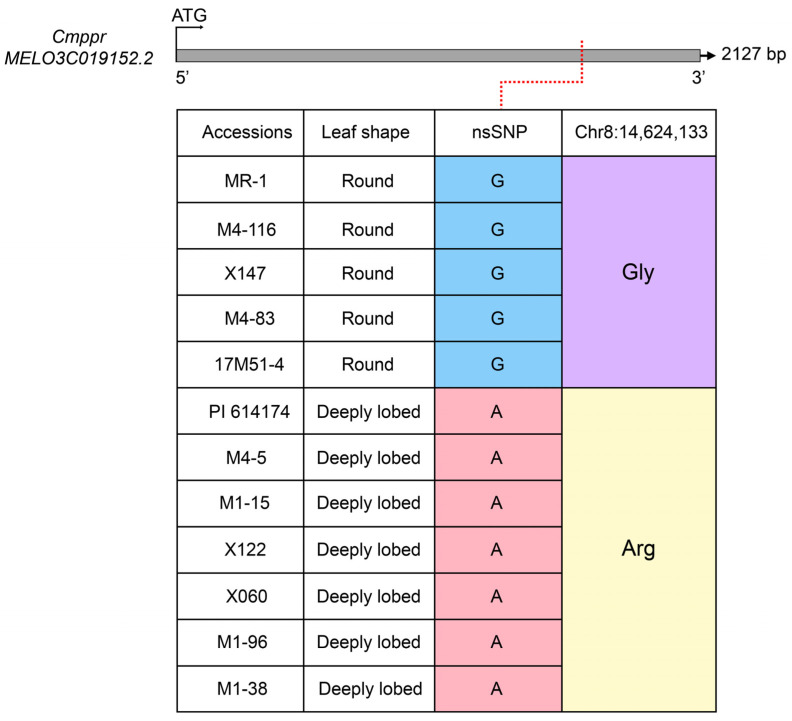
Coding sequence alignment of *MELO3C019152.2* in 12 melon materials including MR-1 and PI 614174.

**Figure 4 plants-13-01134-f004:**
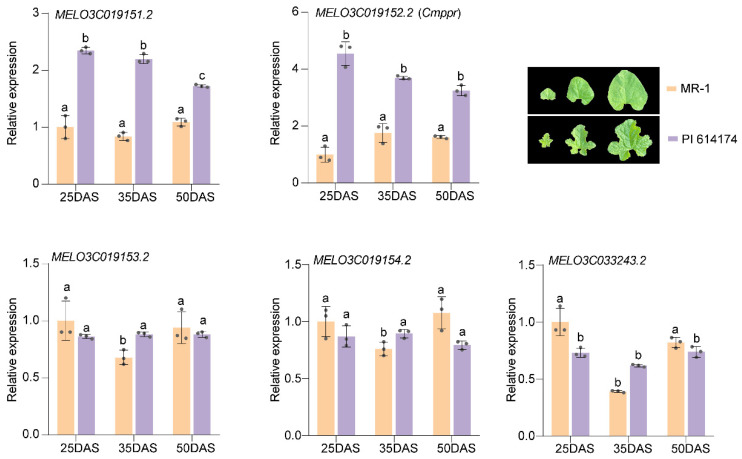
qRT-PCR analysis of five candidate genes within the fine localization interval. The three sets of leaves in the legend represent leaf development at 25, 35 and 50 days after sowing (DAS), respectively. Error bars indicate mean ± standard deviation (SD), and the three data points in each column indicate three biological replicates. Different letters on the figures indicate statistically significant differences (*p* < 0.05).

**Figure 5 plants-13-01134-f005:**
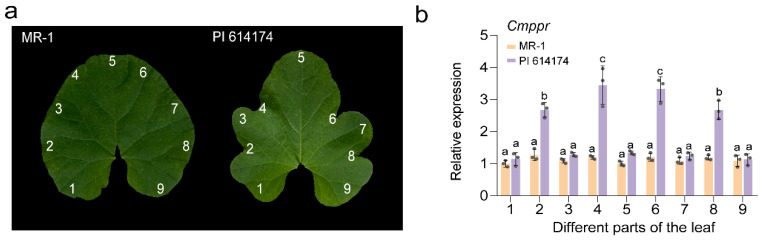
qRT-PCR analysis of *MELO3C019152.2* in nine leaf sites of MR-1 and PI 614174. (**a**) Nine leaf sites of MR-1 and PI 614174. The numbers 1–9 represent nine leaf sites. (**b**) Relative expression of *MELO3C019152.2* in different leaf sites at 50 DAS. Error bars indicate mean ± SD and the three data points in each column indicate three biological replicates. Different letters on the figures indicate statistically significant differences (*p* < 0.05).

**Figure 6 plants-13-01134-f006:**
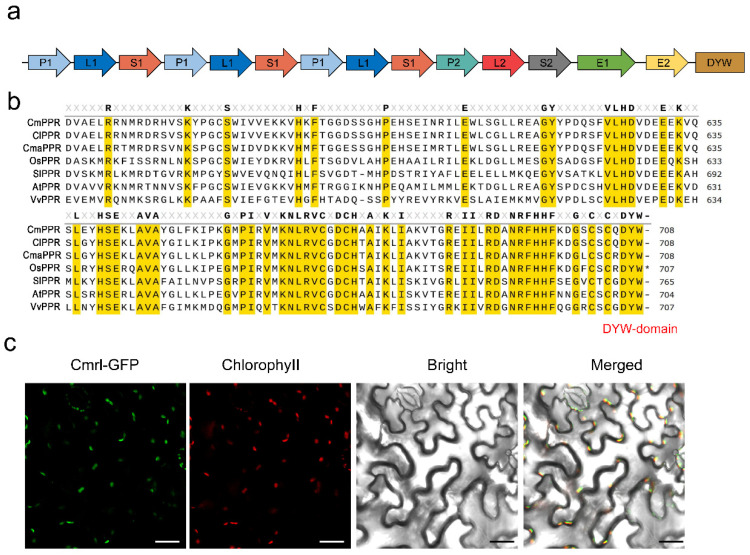
*Cmrl* encodes a DYW-type PPR protein localized in chloroplasts. (**a**) Analysis of the conserved structural domains of *Cmrl*, which contains 15 PPR-repeat motifs and E1, E2 and DYW structural domains at its N-terminus. (**b**) Protein sequence comparison of Cmppr with other plant proximate DYW-type PPR proteins. Red letters indicate DYW conserved domain at the end. Yellow shading represents common conserved areas. The partial PPR protein sequences from melon, watermelon, pumpkin, rice, tomato, *Arabidopsis*, and grape are shown from top to bottom, respectively. (**c**) Subcellular localization of Cmppr in *N. benthamiana* leaves. GFP, green fluorescent protein. Bars = 50 μm.

**Table 1 plants-13-01134-t001:** Segregation of leaf shapes in parents, F_1_ and F_2_ populations from MR-1 and PI 614174.

Population	Plants	Deeply Lobed	Round	Expected Segregation Ratio	Actual Segregation Ratio	*p* Value of *Chi*-Square Tests
MR-1	15	0	15	N/A	N/A	N/A
PI 614174	15	15	0	N/A	N/A	N/A
F_1_	15	15	0	N/A	N/A	N/A
2021-F_2_	220	169	51	3:1	3.31:1	0.648
2021-F_2_	1155	853	302	3:1	2.82:1	0.512

N/A—Not applicable.

## Data Availability

The original contributions presented in the study are included in the article/Appendix A, further inquiries can be directed to the corresponding author/s.

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
