# Peer review of "Identification of Chromosomal Regions and Candidate Genes for *Round leaf* Locus in *Cucumis melo* L."

_plants, 2024, doi:10.3390/plants13081134_

Round 1

Reviewer 1 Report

Comments and Suggestions for Authors

This is an interesting paper regarding dentification of the genes that might control the morphology of melon leaves. In an attempt to clarify their transcriptional regulation, the authors tried to offer genetic resources. Generally the article is well written and I would be inclined to accept it as it stands, but I think the authors should emphasise the relevance of this subject matter to both science and everyday living.

Author Response

Response to Reviewer 1 Comments

  1. This is an interesting paper regarding dentification of the genes that might control the morphology of melon leaves. In an attempt to clarify their transcriptional regulation, the authors tried to offer genetic resources. Generally the article is well written and I would be inclined to accept it as it stands, but I think the authors should emphasise the relevance of this subject matter to both science and everyday living.

Response: Thank you for your recognition of our work and constructive advice. We have added a statement to the Introduction that highlights the significance of the study of leaf morphology in the advancement of the understanding plant phylogeny, vegetation resources and distribution; and in the promotion of biodiversity conservation. In addition, we denote the role of leaf morphological variation in facilitating photosynthesis and respiration, and in adapting to drought and low temperature stressors (Lines 36-38 and Lines 48-55).

Reviewer 2 Report

Comments and Suggestions for Authors

This is a well-written manuscript describing a clean set of well-performed experiments to identify a candidate gene responsible for lobed leaf shape in melon.  My only suggestions are to improve clarity.

1. line 93.  This is the first mention of Cmr1 in the main text.  Please define what Cmr stands for.

2. line 102.  ‘220 F2 populations’ should be either ‘220 F2 individuals’ or ‘F2 population (n=220)’.

3. line 120-122. What sort of reads? Which populations are being used?  F2 for all four; F2 bulks and parents?  Size of the bulks? Clarify for the reader that this is for bulked segregant analysis.  If there are four pools (two F2 populations?) why is there only one delta SNP figure (Fig 2a)? (note that the methods come later, so additional information is needed in the text of the results).

4. Please add significance bars to Fig 2a.

5. line 168.  How large a region was defined as the promoter region?

6. line 170. Why is the TGAC motif interesting?  What is it purported to do?  What is the basis for calling it a ‘cis-acting element’?  in absence of experimental verification, better to say ‘potential cis-acting element’.

7. The lobe specific gene expression is very nice.  At what stage of leaf development was tissue sampled for Fig 5?  Include in text (results and methods) and figure legend.

Author Response

This is a well-written manuscript describing a clean set of well-performed experiments to identify a candidate gene responsible for lobed leaf shape in melon. My only suggestions are to improve clarity.

  1. line 93. This is the first mention of Cmrl in the main text. Please define what Cmrl stands for.

Response 1: Thank you for your valuable comment. We have defined Cmrl in the revised Abstract (Line 21) and in the Introduction section (Line 103).

  1. line 102. ‘220 F2 populations’ should be either ‘220 F2 individuals’ or ‘F2 population (n=220)’.

Response 2: Thank you for your valuable comment. ‘Populations’ has been changed to ‘individuals’ in line 112.

  1. line 120-122. What sort of reads? Which populations are being used? F2 for all four; F2 bulks and parents? Size of the bulks? Clarify for the reader that this is for bulked segregant analysis. If there are four pools (two F2 populations?) why is there only one delta SNP figure (Fig 2a)? (note that the methods come later, so additional information is needed in the text of the results).

Response 3: Thank you for valuable suggestion. The four sets of resequencing data consisted of two parental lines and two extreme pools (round leaf and deeply lobed leaf pools). The two extreme pools were constructed from mixed pools (20 individuals) selected from the F2 population (1,155 individuals), respectively. Detailed data regarding the four resequencing samples have been listed in the revised manuscript (Lines 130-137).

  1. Please add significance bars to Fig 2a.

Response 4: Thank you for calling this issue to our attention. We have added significance bars to Fig 2a. In addition, we have described the significance bars in the Results section (Line 137-139) and figure captions (Line 159-161).

  1. line 168. How large a region was defined as the promoter region?

Response 5: Thank you for your valuable question. The promoter region is 1,919 bp upstream of the start codon ATG of MELO3C019152.2 (Lines 186-187).

  1. line 170. Why is the TGAC motif interesting? What is it purported to do? What is the basis for calling it a ‘cis-acting element’? in absence of experimental verification, better to say ‘potential cis-acting element’.

Response 6: Thank you for your constructive questions and advice. We have changed ‘cis-acting element’ to ‘potential cis-acting element’ in line 190 in response to your insightful suggestion. In our Discussion section (Line 274-282), we discuss in detail the potential TGAC core cis-acting elements with Cmrl. The KNOX-encoded transcription factor acts by binding a core cis-element containing TGAC. In our next study, we will conduct a detailed study of the potential interaction between KNOX and Cmrl.

  1. The lobe specific gene expression is very nice. At what stage of leaf development was tissue sampled for Fig 5? Include in text (results and methods) and figure legend.

Response 7: Thank you for valuable suggestion. The expression results in Fig 5 were obtained from melon leaf tissue at 50 days after sowing. We have added the relevant information to Lines 201, 215, and 346.